# Inactivation of oncogenic cAMP-specific phosphodiesterase 4D by miR-139-5p in response to p53 activation

Bo Cao[1†], Kebing Wang[2†], Jun-Ming Liao[1], Xiang Zhou[1], Peng Liao[1], Shelya X Zeng[1], Meifang He[2], Lianzhou Chen[2], Yulong He[3], Wen Li[2*], Hua Lu[1*]

[1]Department of Biochemistry and Molecular Biology, Tulane Cancer Center, Tulane University School of Medicine, New Orleans, United States; [2]Laboratory of General Surgery, The First Affiliated Hospital, Sun Yat-Sen University, Guangzhou, China; [3]Department of Gastrointestinal Surgery, The First Affiliated Hospital, Sun Yat-sen University, Guangzhou, China

**Abstract** Increasing evidence highlights the important roles of microRNAs in mediating p53's tumor suppression functions. Here, we report miR-139-5p as another new p53 microRNA target. p53 induced the transcription of miR-139-5p, which in turn suppressed the protein levels of phosphodiesterase 4D (PDE4D), an oncogenic protein involved in multiple tumor promoting processes. Knockdown of p53 reversed these effects. Also, overexpression of miR-139-5p decreased PDE4D levels and increased cellular cAMP levels, leading to BIM-mediated cell growth arrest. Furthermore, our analysis of human colorectal tumor specimens revealed significant inverse correlation between the expression of miR-139-5p and that of PDE4D. Finally, overexpression of miR-139-5p suppressed the growth of xenograft tumors, accompanied by decrease in PDE4D and increase in BIM. These results demonstrate that p53 inactivates oncogenic PDE4D by inducing the expression of miR-139-5p.

*For correspondence: wenli28@163.com (WL); hlu2@tulane.edu (HL)

[†]These authors contributed equally to this work

Competing interests: The authors declare that no competing interests exist.

## Introduction

microRNAs (miRNAs) represent a class of cellular short non-coding RNAs responsible for modulating the expression of their target genes at the post-transcriptional level. Abnormal regulation of miRNAs is associated with human cancer development and staging (*Lu et al., 2005*). Over the past decade, increasing attention has been drawn to the role of miRNAs in p53 signaling network, and a number of miRNAs have been identified as p53 target genes (*Liao et al., 2014a*). These miRNAs are involved in multiple biological processes, including cell cycle arrest, apoptosis, glycolysis and so on. Also, they often connect p53 with other signaling pathways (*Christoffersen et al., 2010*; *Sachdeva et al., 2009*; *Liang et al., 2013*). Although miRNAs have been appreciated as important mediators of p53's tumor suppression functions, a lot still remain unexplored to better understand the fine-tuning of p53 signaling and crosstalk with other pathways by these RNAs. In this study, we identified miR-139-5p as a novel p53 target gene and demonstrated a new pathway connecting p53 and miR-139-5p with an oncogenic protein PDE4D as a new target of this miRNA and its downstream cAMP signaling.

## Results and discussion

### Identification of miR-139 as a novel p53 target gene

In order to identify potential p53 target microRNAs, we used colon cancer cell lines, HCT116 p53[+/+] and HCT116 p53[-/-], the latter of which were genetically engineered to lose the expression of wild

**eLife digest** The human body is kept mostly free from tumors by the actions of so-called tumor suppressor genes. One such gene encodes a protein called p53, which prevents tumors from growing by regulating the activity of many other genes that either inhibit cell growth or cause cells to die. For example, p53 regulates genes that encode short molecules called microRNAs, which in turn suppress the activity of other target genes.

Although a number of microRNAs have been reported as p53-regulated genes, there are still more to find. Discovering these genes would in turn help researchers to better understand exactly how p53 acts to suppress the growth of tumors, and to treat cancers caused by mutations in this tumor suppressor gene.

Cao, Wang et al. now discover a new microRNA – called miR-139-5p – as one that is activated by p53 in human cells. Colon tumors produce much lower levels of this microRNA than normal tissues, while the cancer cells with a higher level of miR-139-5p grow slower than do the cancer cells with less miR-139-5p. Further experiments showed that this is because miR-139-5p can suppress the production of a protein called PDE4D, which is often highly expressed in human cancers. The suppression of PDE4D by this microRNA results in an increase in the levels of a protein that can cause cancer cells to die.

Cao, Wang et al. suggest that miR-139-5p and PDE4D form part of a signaling pathway that plays an important role in suppressing the growth of colon cancer cells. Since microRNAs often have more than one target, future studies could explore if miR-139-5p regulates the production of other cancer-related proteins as well.

type (WT) p53 (*Bunz et al., 1998*). Both cells were treated with 4 μM Inauhzin-C (INZ-C), which is a p53 activating compound discovered in our lab (*Zhang et al., 2012*). After confirmation of p53 and its targets induction through Western blot (*Figure 1A*) and quantitative real-time PCR (qRT-PCR) (*Figure 1B*), the total RNA was extracted and sent to ArrayStar for miRNA-sequencing analysis. The results revealed that in addition to known p53 target microRNAs, such as miR-34a, miR-1246 and miR-143 (*Liao et al., 2014a*; *Suzuki et al., 2009*), miR-139 was also significantly induced in HCT116 p53$^{+/+}$, but not HCT116 p53$^{-/-}$, cells, suggesting miR-139 as a potential p53 target (*Figure 1C*). We independently confirmed this observation by detecting miR-139-5p expression after treating HCT116 p53$^{+/+}$/HCT116 p53$^{-/-}$ and H460 (WT p53)/H1299 (p53 null) cells with DMSO or 4 μM INZ-C. miR-139-5p was significantly induced only in p53 positive, but not in p53 null, cells (*Figure 1D*). In contrast, p53 knockdown decreased miR-139-5p expression by more than 50% in H460 and U2OS cells (*Figure 1E*). These data indicate that miR-139-5p is a possible p53 target gene.

## miR-139 is a direct p53 target gene

After carefully analyzing the genomic sequence of the miR-139 gene using p53MH algorism (*Hoh et al., 2002*), we found a highly conserved putative p53 responsive element (RE) located at the 5' flanking region (*Figure 2A*). To test if endogenous p53 binds to this p53RE sequence, we conducted a chromatin-associated immunoprecipitation (ChIP) assay after treating H460 or HCT116 p53$^{+/+}$ cells with 0.5 μM Doxorubicin for p53 induction. In both cell lines, the binding of p53 with the p53RE was dramatically increased upon Doxorubicin treatment as compared to non-treatment control, indicated by p53RE sequence pulled down with p53 specific antibody DO-1, but not non-specific immunoglobulin G (*Figure 2B*). Also, we assessed luciferase expression driven by either a wild type or a mutant-p53RE-motif-containing miR-139 sequence (*Figure 2C*) in H1299 cells. GFP-p53 markedly induced luciferase activity in a dose-dependent manner when wild type miR-139 p53RE, but not the mutant p53RE, was used (*Figure 2D*). These results clearly show that p53 binds to the miR-139 promoter region and thus regulates the transcription of miR-139.

## p53 regulates PDE4D via miR-139-5p

By using the online microRNA target prediction tool (*Maragkakis et al., 2009*), we searched for potential RNA targets of miR-139-5p. After screening several candidates, PDE4D turned out to be

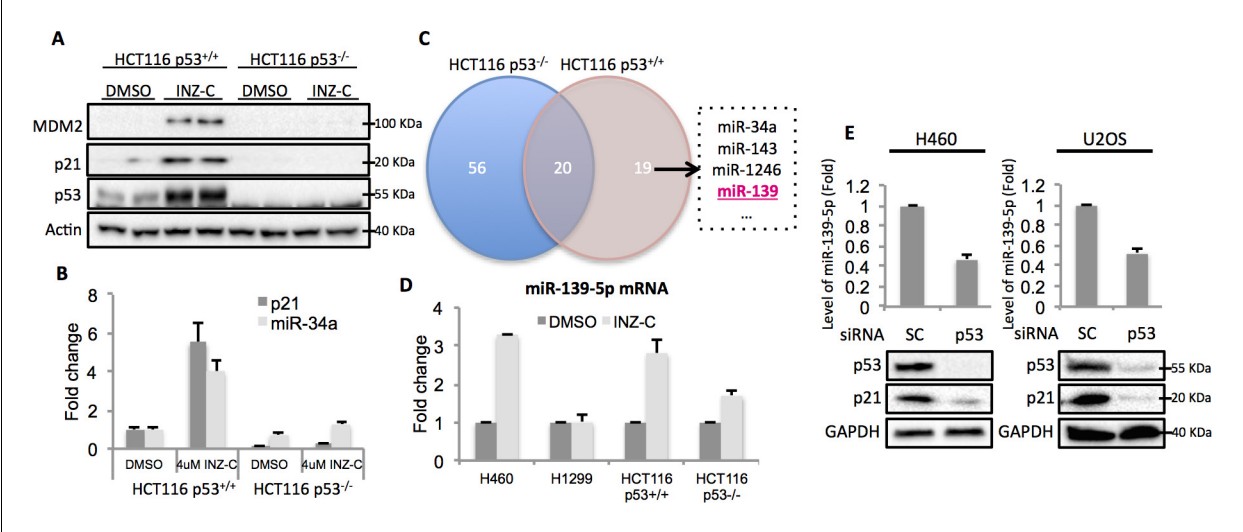

**Figure 1.** Identification of miR-139 as a novel p53-responsive gene. Validation of induction of p53 and its targets gene. HCT116 p53+/+ and HCT116 p53-/- cells were treated with DMSO or 4uM INZ-C for 18 hr in duplicate and the protein samples and RNA samples were prepared for Western Blot (A) and qRT-PCR (B), respectively. (C). microRNA sequencing data analysis shows significantly induced microRNAs uniquely or commonly in HCT116 p53-/- and HCT116 p53+/+ cells. miR-139 was highlighted in red among other known p53 targets. (D). Induction of miR-139-5p by p53. p53 positive (H460, HCT116 p53$^{+/+}$) and p53 negative (H1299, HCT116 p53$^{-/-}$) cells were treated as in (A) and (B), and qRT-PCR analysis followed. (E). Decrease of miR-139-5p by knocking down p53. H460 and U2OS cells were transfected with scramble siRNA (SC) or siRNA specific to p53 (p53) and subjected to Western blot and qRT-PCR analysis.

an ideal target as its 3'-untranslated region (3'-UTR) contains miR-139-5p targeting sequence (*Figure 3A*). Overexpression of miR-139-5p mimic markedly reduced the expression of PDE4D in H460 and A549 cells (*Figure 3B*). The type-I insulin-like growth factor receptor (IGF-IR), a previously reported miR-139-5p target (*Shen et al., 2012*), was also decreased by miR-139-5p mimic, indicating the activity of miR-139-5p used here (*Figure 3—figure supplement 1*). PDE4D belongs to the family of phosphodiesterases and is a cyclic AMP (cAMP) specific phosphodiesterase with several splice variants (*Omori and Kotera, 2007*). It is an oncogenic protein that regulates cancer cell proliferation, angiogenesis and apoptosis (*Lin et al., 2013*; *Ogawa et al., 2002*; *Pullamsetti et al., 2013*; *Rahrmann et al., 2009*). To further test whether the downregulation of PDE4D by miR-139-5p is through direct regulation on its mRNA, we constructed the WT or mutant predicted miR-139-5p target sites into the pMIR-Report system that contains the luciferase reporter gene subjected to regulation mimicking the microRNA target (*Figure 3C*). When co-transfecting H1299 cells with miR-139-5p, only the pMIR-PDE4D-WT, but not the pMIR-PDE4D-mutant, displayed suppressed expression (*Figure 3D*), suggesting that miR-139-5p regulates PDE4D expression by directly binding to the target sequence at 3'-UTR of PDE4D mRNA.

In line with the above results, activation of p53 by doxorubicin in H460 and A549 cells significantly reduced the protein levels of PDE4D (*Figure 3E*, left panel). Notably, the PDE4D variants with smaller molecular weight have been reported to possess stronger oncogenic activities due to the lack of functional inhibitory domains as compared to the longer forms (*Lin et al., 2013*). Nevertheless, miR-139-5p transfection and doxorubicin treatment led to similar expression inhibition of both the long and the short variants of PDE4D, indicating that this newly identified pathway has broader inhibition on PDE4D. In contrast, knocking down p53 elevated PDE4D expression in these two cell lines (*Figure 3E*, right panel). We then assessed the effect of miR-139-5p on PDE4D in PC-3, a p53 null cell line, and found similar reduction of PDE4D by miR-139-5p mimic (*Figure 3—figure supplement 2*). To further validate the regulatory role of p53 on PDE4D, we also treated HCT116 p53+/+ and HCT116 p53-/- cells with 10 µM Nutlin-3 (Nut), which disrupts MDM2-p53 interaction and therefore activates p53 (*Vassilev et al., 2004*), and 5 nM Actinomycin D (ActD), which causes ribosomal stress-mediated p53 activation (*Iapalucci-Espinoza and Franze-Fernández, 1979*), and found PDE4D was reduced by both of these two drugs only in p53 positive, but not p53 negative, cells

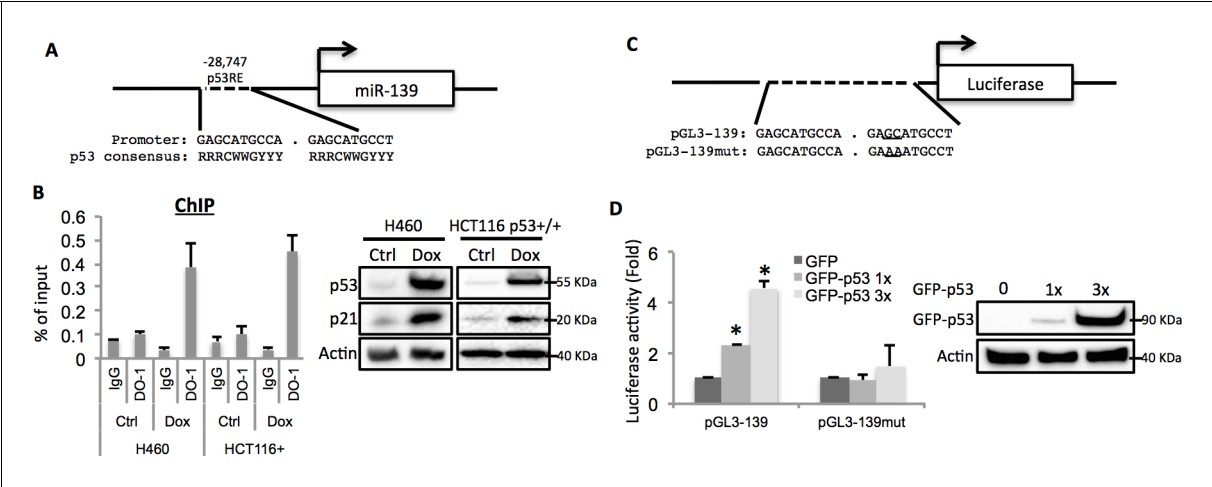

**Figure 2.** miR-139 is a direct target of p53. (**A**) Diagram shows putative p53 responsive element (p53RE) located upstream of miR-139 gene. (**B**) Increased binding of p53 and the endogenous p53RE-containing miR-139 promoter in response to Doxorubicin. H460 or HCT116 p53+/+ cells were treated with 0.5μM Doxorubicin for 18 hr to stimulate the endogenous p53 before chromatin-associated immunoprecipitation assays were conducted with DO-1 p53 antibodies and the p53RE specific primers listed in *Supplementary file 1* online. Western blot analysis was performed to confirm induction of p53. (**C**) Schematic of the pGL3 luciferase reporter constructs used. The plasmids either contain wild-type or mutant p53RE sequences of the miR-139 promoter, as underlined. (**D**). Enhancement of miR-139-promoter-driven luciferase activity by p53. H1299 cells were co-transfected with pGL3-139 or pGL3-139mut and increasing amount of GFP-p53 and collected 48 hr after transfection for assessment of luciferase activity, which was normalized against β-gal expression. Western blot was also performed to confirm the expression of GFP-p53. Error bars represent standard deviation (n = 3). β-gal, β-galactosidase; Ctrl, control; Dox, Doxorubicin; IgG, immunoglobulin G; mut, mutant.

(*Figure 3—figure supplement 3*). These results suggest that this suppression of PDE4D is p53 dependent in response to various stresses.

Furthermore, doxorubicin inhibited pMIR-PDE4D activity, but introduction of miR-139-5p inhibitor reversed this inhibition (*Figure 3F*). Consistently, the suppression of PDE4D protein by doxorubicin was also compromised in the presence of miR-139-5p inhibitor (*Figure 3G*). Collectively, these findings demonstrate that activation of p53 can induce the expression of miR-139-5p that in turn suppresses the expression of oncogenic protein PDE4D.

## miR-139-5p induces cellular cAMP and BIM-mediated cell growth arrest

Since PDE4D is a cAMP specific phosphodiesterase, ectopic expression of miR-139-5p in A549 cells led to significant increase of cellular cAMP levels (*Figure 4A*). Also, consistent with our results shown in *Figure 3F* and *Figure 3G*, doxorubicin treatment increased cellular cAMP levels in A549 cells, which were alleviated by the miR-139-5p inhibitor (*Figure 4—figure supplement 1*). In addition, Nutlin-3, a more specific p53-activating reagent, showed similar effect on cAMP level, which was also reversed in the presence of miR-139-5p inhibitor (*Figure 4B*). In contrast, in H1299 cells, which are p53-null and express non-detectable PDE4D, neither doxorubicin nor miR-139-5p affected cellular cAMP level (*Figure 4—figure supplement 2*). Notably, the rescue effect of miR-139-5p on A549 cell growth inhibition by Nutlin-3 was correlated with the change of cAMP levels (*Figure 4—figure supplement 3*). These data indicate that p53 activation could modulate cAMP levels through miR-139-5p.

Depletion of PDE4D was previously reported to induce BIM-mediated apoptosis through activation of the cAMP pathway (*Lin et al., 2013*; *Zambon et al., 2011*). We found that introduction of miR-139-5p into H460 and A549 cells dramatically increased BIM protein expression (*Figure 4C*). As expected, treatment with 10 μM Nutlin-3 also induced BIM expression in HCT116 p53[+/+], but not HCT116 p53[-/-], cells (*Figure 4—figure supplement 4*), supporting that BIM induction is p53-dependent. Consequently, A549 cell growth was significantly inhibited by miR-139-5p or Nutlin-3, both of which were attenuated by knocking down BIM using siRNA (*Figure 4D*, *Figure 4—figure supplement 5*). Inversely, introduction of miR-139-5p inhibitor into A549 cells significantly alleviated cell growth inhibition by several p53 activating drugs, including INZ-C, Actinomycin D and Nutlin-3

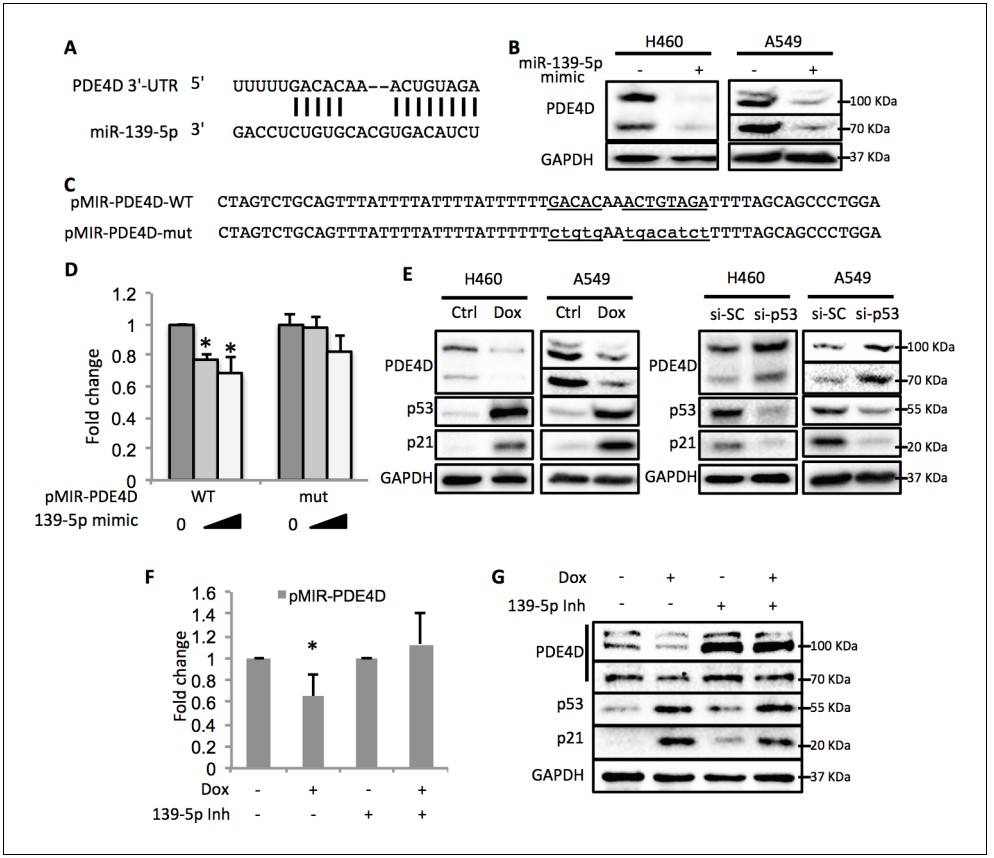

**Figure 3.** p53 modulates PDE4D expression via miR-139-5p. (**A**) Bioinformatic analysis shows the miR-139-5p-targeted 3'-untranslated region (3'UTR) sequence of PDE4D mRNA. (**B**) Overexpression of miR-139-5p decreases the level of PDE4D protein in cells. H460 and A549 cells were collected 48 hr after transfection with miR-139-5p mimic for Western blot analysis. (**C**) Schematic of the pMIR–PDE4D luciferase reporter constructs used, which contain either a wild-type or a mutant miR-139-5p target site derived from the PDE4D mRNA. (**D**) Overexpression of miR-139-5p specifically inhibits luciferase activity from the plasmid harboring a wild-type, but not a mutant, miR-139-5p targeted sequence. H1299 cells were co-transfected with the indicated plasmids and collected 48 hr after transfection for luciferase assay. Luciferase activity was measured and normalized against β-gal expression. Even amount of oligos was achieved by adding Negative control oligos accordingly. (**E**) p53 modulates PDE4D expression. H460 and A549 cells were collected for Western blot analysis 18 hr after treatment with Doxorubicin (left panel) or 48 after transfection with si-SC or si-p53 (right panel). (**F** and **G**) p53 modulation of PDE4D expression is through miR-139-5p. H460 cells were co-transfected with indicated plasmids and 24 hr later were treated with Doxorubicin for 18 hr followed by measurement of luciferase activity (**F**) or Western blot analysis (**G**). 139-5p m, miR-139-5p mimic; 139-5p Inh, miR-139-5p inhibitor. Cells were treated with solvent for Dox or transfected with the same concentration of negative control oligos as miR-139-5p mimic or inhibitor. Error bars represent standard deviation (n = 3). *p<0.05.

The following figure supplements are available for figure 3:

**Figure supplement 1.** HepG2 cells were collected 48 hr after transfection with miR-139-5p mimic at 0, 20 or 40 nM for Western blot analysis on IGF-IR expression.

**Figure supplement 2.** PC-3 cells were collected 48 hr after transfection with miR-139-5p mimic at 40 nM for Western blot analysis on PDE4D expression.

**Figure supplement 3.** HCT116 p53[+/+] and HCT116 p53[-/-] cells were treated with 10 μM Nutlin-3 (Nut) or 5 nM Actinomycin D (ActD) for 18 hr, and then collected for Western blot analysis.

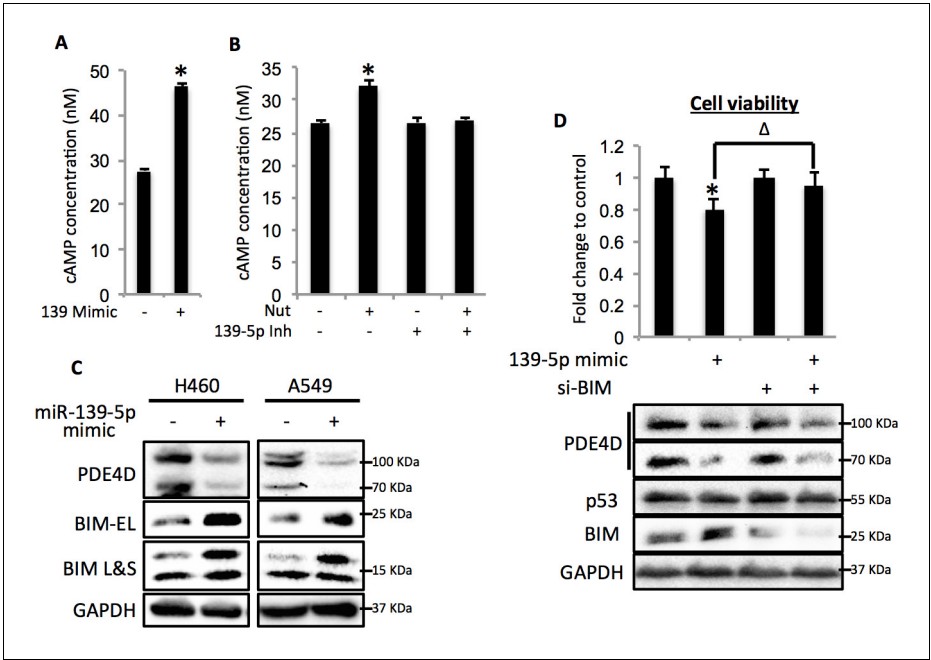

**Figure 4.** miR-139-5p induces cAMP/BIM mediated cell growth arrest. (**A**). Overexpression of miR-139-5p increases cellular cAMP levels. A549 cells were transfected with miR-139-5p mimic control or miR-139-5p mimic and were subjected to cAMP measurement 48 hr after transfection. (**B**). Nutlin-3 increases cellular cAMP levels via miR-139-5p. A549 cells were transfected with miR-139-5p inhibitor control or miR-139-5p inhibitor for 48 hr followed by 10 μM Nutlin-3 treatment for 18 hr and subjected to cAMP measurement. (**C**). Overexpression of miR-139-5p induces BIM expression. H460 and A549 cells were transfected with miR-139-5p mimic and Western blot performed 48 hr after transfection. (**D**). Knockdown of BIM rescues miR-139-5p induced cell growth arrest. A549 cells were transfected with miR-139-5p mimic or siRNA against BIM (si-BIM) or both and subjected to MTT assay 48 hr after transfection. Cells were treated with solvent for Nutlin-3 or transfected with the same concentration of negative control oligos as miR-139-5p mimic or BIM siRNA. Error bars represent standard deviation (n = 3). *p<0.05 as compared to negative control oligos. Δ, p<0.05.

The following figure supplements are available for figure 4:

**Figure supplement 1.** Doxorubicin increases cellular cAMP levels via miR-139-5p.

**Figure supplement 2.** H1299 cells were treated with vehicle or 0.5 μM Doxorubicin for 18 hr followed by cAMP measurement, or transfected with miR-139-5p mimic control or miR-139-5p mimic and subjected to cAMP measurement 48 hr after transfection.

**Figure supplement 3.** miR-139-5p inhibitor rescues Nutlin-3 inhibition of cell growth.

**Figure supplement 4.** HCT116 p53+/+ and HCT116 p53-/- cells were treated with vehicle, or 10 μM Nutlin-3 for 18 hr, and subjected to Western blot analysis.

**Figure supplement 5.** BIM knockdown alleviates Nutlin-3 inhibition of cell growth.

**Figure supplement 6.** A549 cells were transfected with miRNA inhibitor control or miR-139-5p inhibitor, and 48 hr after transfection, the cells were treated with vehicle, 2 μM INZ-C, 5 nM ActD or 10 μM Nutlin-3 for 48 hr, followed by MTT assay to determine cell viability.

(*Figure 4—figure supplements 3* and *6*). These results demonstrate that in response to p53 activation, increased miR-139-5p induces BIM-mediated cell growth arrest *via* the PDE4D/cAMP pathway.

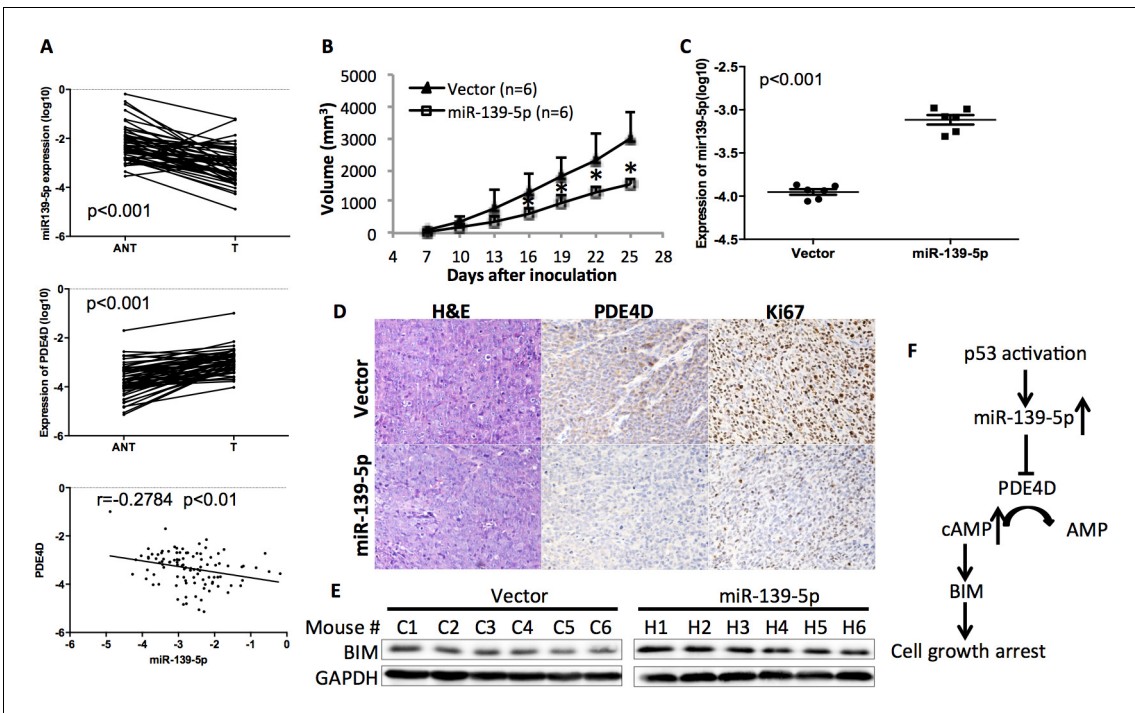

**Figure 5.** miR-139-5p is negatively correlated with PDE4D expression in human colorectal tumor samples, and represses the growth of SW480 xenograft tumors. (**A**) miR-139-5p is negatively correlated with PDE4D expression in colon tumor samples. miR-139-5p (top panel) and PDE4D (middle panel) RNA expression was determined in 50 tumors (T) and paired adjacent normal tissues (ANT). The Pearson's correlation of miR-139-5p and PDE4D RNA expression was analyzed combining tumor samples and normal tissues (bottom panel). (**B**) The sizes of SW480 xenograft tumors stably overexpressing pEZX-control (Vector) or pEZX-miR-139-5p (miR-139-5p) were measured every three days starting at seven days after inoculation. Mean tumor sizes were presented. Error bar, SD; *p<0.05. (**C**) Expression of miR-139-5p of xenograft tumors was determined by qRT-PCR. (**D**) Representative images of the xenograft tumors stained with hematoxylin and eosin (H&E), or immunohistochemistry analyzed with PDE4D or Ki67 antibody. (**E**) Xenograft tumors were subjected to Western blot analysis of BIM expression. (**F**) Proposed model of the p53/miR-139-5p/PDE4D pathway.

The following figure supplement is available for figure 5:

**Figure supplement 1.** Quantification of PDE4D and Ki67 IHC analysis.

## miR-139-5p is negatively correlated with PDE4D in human and xenograft colorectal tumors

In order to determine the clinical relevance of miR-139-5p regulation of PDE4D, we obtained 50 paired human colon tumor specimens and their adjacent normal tissues and conducted qRT-PCR analysis on these specimens. miR-139-5p expression was significantly lower while PDE4D was higher in these tumor specimens than that in their normal tissues (*Figure 5A*). Pearson's correlation analysis of the expression results from the tumor specimens and normal tissues revealed that miR-139-5p is inversely correlated with PDE4D expression (*Figure 5A*, bottom panel). These results provide clinical evidence supporting miR-139-5p as a negative regulator of PDE4D, consistent with our above results.

To validate this clinical correlation, we established a xenograft model using SW480 cells stably expressing either scramble oligos (Control) or miR-139-5p. As expected, miR-139-5p expressing tumors grew significantly slowly as compared to control tumors starting at day 16 after inoculation (*Figure 5B*). The difference of miR-139-5p expression in these two groups of tumors was comparable to that observed in human specimens (*Figure 5C* vs *Figure 5A*). Immunohistochemistry analysis revealed that in the miR-139-5p overexpressing tumors, PDE4D expression was markedly repressed, while tumor cell proliferation was significantly inhibited as reflected by Ki67 staining (*Figure 5D* and

Figure 5—figure supplement 1). Consistent with our in vitro observation, BIM expression was also elevated in miR-139-5p tumors (*Figure 5E*). These findings suggest that the tumor suppressor role of miR-139-5p is likely attributed to its regulation of the PDE4D/BIM pathway.

In summary, this study for the first time demonstrates that p53 can induce the expression of miR-139-5p (*Figure 1* and *Figure 2*), which in turn suppresses the expression of an oncogenic protein PDE4D (*Figure 3*) and leads to cAMP/BIM-mediated cell growth arrest (*Figure 4*). Significantly, miR-139-5p expression is negatively correlated with PDE4D in human colorectal tumor and normal tissues, and overexpression of miR-139-5p is associated with slower tumor growth in the xenograft model, which is accompanied with PDE4D suppression, BIM induction and cell proliferation inhibition (*Figure 5*). As a potential tumor suppressor, miR-139 was previously shown to be downregulated in human hepatocellular carcinoma and colorectal cancer with characterized targets including Rho-kinase 2, IGF-IR and RAP1B (*Guo et al., 2012*; *Shen et al., 2012*; *Wong et al., 2011*). Here, we identified PDE4D, an oncogenic protein that is upregulated in various human cancers (*Lin et al., 2013*), as another novel target of this miRNA. Inhibition or depletion of PDE4D significantly induces apoptosis and inhibits proliferation of cancer cells (*Lin et al., 2013*; *Ogawa et al., 2002*; *Rahrmann et al., 2009*). Notably, the oncogenic property of PDE4D involves the cAMP/BIM pathway (*Lin et al., 2013*; *Zambon et al., 2011*). cAMP is an important secondary messenger mediating diverse cellular processes with protein kinase A as its main effector (*Taskén and Aandahl, 2004*). In particular, lower cAMP levels favor cancer cell survival and proliferation, which can be reversed by inhibition of PDE4D, the cAMP specific phopsphodiesterase (*Goldhoff et al., 2008*; *Lin et al., 2013*; *Murata et al., 2000*; *Ogawa et al., 2002*). The tumor suppressor role of p53 has been extensively documented over the last two decades, and is highly attributable to its regulation of target genes involved in cell cycle arrest, apoptosis and senescence (*Bieging et al., 2014*; *Levine, 2011*). More recent discoveries revealed that p53 is also a critical mediator of metabolic pathways that are important for tumor survival (*Bieging et al., 2014*; *Jiang et al., 2015*; *Wang and Gu, 2014*). Based on our findings, we propose a p53-miR-139-5p-PDE4D-cAMP-BIM pathway as a novel pathway that can mediate p53's tumor suppression function to modulate cellular cAMP levels by inhibiting PDE4D expression via miR-139-5p, and deregulation of this pathway would be highly associated with tumorigenesis (*Figure 5F*).

## Materials and methods

### Cell lines

Human HCT116 p53$^{+/+}$ (RRID: CVCL_0291) and HCT116 p53$^{-/-}$ (RRID: CVCL_S744) cells were generous gifts from Dr. Bert Vogelstein at the John Hopkins Medical Institutes. A549 (RRID: CVCL_0023), HepG2 (RRID: CVCL_0027), U2OS (RRID: CVCL_0042), H460 (RRID: CVCL_0459) and H1299 (RRID: CVCL_0060) cells were purchased from American Type Culture Collection (ATCC). All cells were cultured in Dulbecco's modified Eagle's medium (DMEM) and PC3 (RRID: CVCL_0035) cells (also from ATCC) in RPMI 1640 medium supplemented with 10% fetal bovine serum, 50 U/ml penicillin and 0.1 mg/ml streptomycin at 37°C in 5% $CO_2$. STR profiling was performed to ensure cell identity. No mycoplasma contamination was found.

### Plasmids and antibodies

The pGL3-miR-139 luciferase reporter plasmid was constructed from the miR-139 promoter with primers as listed in *Supplementary file 1*. The fragment was inserted into pGL3 at the MluI and XhoI sites. The pGL3-miR-139-mut was generated by site-directed mutagenesis with primers listed in *Supplementary file 1* using pGL3-miR-139 as template. The pMIR-PDE4D and pMIR-PDE4D mutant plasmids were constructed by inserting the miR-139-5p-targeted PDE4D mRNA-coding sequence or its mutant into the pMIR vector (Ambion) at the SpeI and HindIII sites. The pSIF-H1-miR-139-5p was constructed by inserting annealed oligos as listed in *Supplementary file 1* into the pSIF-H1 vector at BamHI and EcoRI sites, as per manufacturer's instruction. The anti-p21 (Thermo Fisher Scientific, Waltham, MA, RRID: AB_10986834), anti-p53 (DO-1, Santa Cruz, Dallas, TX, RRID: AB_628082), anti-PDE4D (Aviva Systems Biology, San Diego, CA, RRID: AB_10879817), and anti-BIM (Cell Signaling, Danvers, MA, RRID: AB_10692515) antibodies used here were commercially purchased. The anti-MDM2 (2A10 and 4B11) antibodies were described previously (*Jin et al., 2002*).

## Chemicals

Inauhzin-C (INZ-C) was reported previously (*Zhang et al., 2012*). Doxorubicin (Dox) was purchased from Sigma-Aldrich.

## Quantitative real-time PCR (qRT-PCR)

qRT-PCR for mature microRNAs was carried out by using the methods as described previously (*Tang et al., 2006*). qRT-PCR for other genes were conducted using the protocol as described before (*Sun et al., 2008*). Relative gene expression was calculated using the ΔΔCT method. All reactions were carried out in triplicate.

## Transient transfection and immunoblot

As described previously (*Jin et al., 2008*), briefly, cells were transfected with plasmids as indicated in each figure by using TurboFect (Thermo Scientific, Waltham, MA) and following the company's manuals. Cells were harvested and lysed in lysis buffer 48 hr post transfection. The total protein concentrations were determined using a BioRad protein assay kit and equal amounts of total proteins (50 μg, otherwise indicated specifically) were then subjected to SDS-PAGE, followed by WB with antibodies as indicated in each figure.

## Knockdown of the endogenous miRNA, p53 and BIM

Hsa-mir-139-5p mimic and Negative control were purchased from Gene Pharma (Shanghai, China). Anti-miR miRNA Inhibitor and Anti-miR miRNA Inhibitors—Negative Control were purchased from Ambion. p53 siRNA was purchased from Santa Cruz. Two BIM siRNA (siBIM-1, ID: 19,474 and siBIM-2, ID: 195012) were purchased from Ambion. Transfection of miRNA inhibitors was performed using the same method as that for normal siRNA as described previously (*Sun et al., 2008*).

## Luciferase reporter assays

Cells were transfected with pCMV-β-galactoside and indicated plasmids (total plasmid DNA 1 μg/well) as indicated in figures. Luciferase activity was determined and normalized by a factor of β-gal activity in the same assay as described previously (*Jin et al., 2006*).

## ChIP–PCR

ChIP analysis was performed as described previously using p53 (DO-1) antibodies for endogenous p53 (*Liao and Lu, 2013*; *Liao et al., 2014b*). Immunoprecipitated DNA fragments were analyzed by quantitative real-time PCR (qRT-PCR) amplification using primers for miR-139 gene. The primers are listed in *Supplementary file 1* online.

## Cellular cAMP measurement

cAMP-GloTM Assay (Promega, Madison, WI) was performed to measure the cellular cAMP concentration as per manufacturer's instructions.

## Tissue specimens

Fifty paired colorectal tumors and adjacent non-tumor tissues were collected and histopathologically diagnosed at the Departments of Gastrointestinal Surgery and Pathology, the First Affiliated Hospital, Sun Yat-sen University. Patient consent and Institutional Research Ethics Committee approval were obtained prior to the use of these clinical materials for research purposes.

## Xenograft

SW480 cells ($1 \times 10^7$) stably expressing pEZX-scramble control sequence (Vector) or pEZX-miR-139-5p were inoculated subcutaneously into the right flank of female BALB/c nude mouse (four weeks old, n = 6 per group). The tumor volume was measured every three days and calculated as $0.524 \times$ length $\times$ width$^2$ (*Gleave et al., 1992*). At the conclusion of the experiments, tumors were removed and fixed in 10% formalin for paraffin embedding and histological analysis, or flash-frozen in liquid nitrogen for Western blot and qRT-PCR analysis. All experimental procedures were approved by the Medical Ethical Committee of the First Affiliated Hospital, Sun Yat-sen University (Guangzhou, China). H & E staining and immunohistochemistry were described previously (*Cao et al., 2013*;

*Zhang et al., 2013*). Quantitative analysis of IHC staining was achieved by categorizing the staining intensity to low, medium and high as determined by ImageJ software (NIH).

## Statistical analysis

The Student's two-tailed *t* test was used to compare the mean differences between treatment and control groups, unless otherwise indicated. Data are presented as Mean ± SD (standard deviation). $p < 0.05$ was determined as statistically significant.

## Acknowledgements

Hua Lu was supported in part by NIH-NCI grants R01CA095441, R01CA172468, R01CA127724, and R21 CA190775 as well as the Reynolds and Ryan Families Chair fund. Wen Li was supported by the National Natural Science Foundation of China (No. 81172337, 30973395).

## Additional information

### Funding

| Funder | Grant reference number | Author |
| --- | --- | --- |
| National Natural Science Foundation of China | 81172337 | Wen Li |
| National Natural Science Foundation of China | 30973395 | Wen Li |
| NIH Office of the Director | R01CA095441 | Hua Lu |
| NIH Office of the Director | R01CA172468 | Hua Lu |
| NIH Office of the Director | R01CA127724 | Hua Lu |
| NIH Office of the Director | R21 CA190775 | Hua Lu |

The funders had no role in study design, data collection and interpretation, or the decision to submit the work for publication.

### Author contributions

BC, Conception and design, Acquisition of data, Analysis and interpretation of data, Drafting or revising the article; KW, Conception and design, Acquisition of data, Analysis and interpretation of data; J-ML, XZ, PL, SXZ, Conception and design, Acquisition of data; MH, LC, YH, Acquisition of data; WL, HL, Conception and design, Analysis and interpretation of data, Drafting or revising the article

### Author ORCIDs

Hua Lu, http://orcid.org/0000-0002-9285-7209

### Ethics

Animal experimentation: This study was performed in strict accordance with the recommendations in the Guide for the Care and Use of Laboratory Animals of the National Institutes of Health. All of the animals were handled according to approved institutional animal care and use committee (IACUC) protocols (#4257R) of Tulane University. All surgery was performed under sodium pentobarbital anesthesia, and every effort was made to minimize suffering.

## Additional files

### Supplementary files

• Supplementary file 1. List of primers used in this study.

## Major datasets

The following dataset was generated:

| Author(s) | Year | Dataset title | Dataset URL | Database, license, and accessibility information |
|---|---|---|---|---|
| Bo Cao, Hua Lu | 2016 | Inactivation of Oncogenic cAMP-specific Phosphodiesterase 4D by miR-139-5p in Response to p53 Activation | http://www.ncbi.nlm.nih.gov/geo/query/acc.cgi?acc=GSE79099 | Publicly available at the NCBI Gene Expression Omnibus (Accession no: GSE79099) |

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
