## [Decision Letter]

[Editors’ note: a previous version of this study was rejected after two rounds of review, but the authors submitted for reconsideration. The first decision letter after peer review is shown below.]

Thank you for sending your work entitled "Inactivation of Oncogenic cAMP-specific Phosphodiesterase 4D by miR-139-5p in Response to p53 Activation" for consideration at *eLife*. Your article has been evaluated by a Senior editor and three reviewers, one of whom is a member of our Board of Reviewing Editors. The reviewers found the work potentially quite interesting, but raised a large number of concerns, all of which would need to be addressed successfully to make the paper suitable, after re-review, for *eLife*.

The Reviewing editor and the other reviewers discussed their comments before we reached this decision, and the Reviewing editor has assembled the following comments to help you prepare a revised submission.

The authors performed a differential expression analysis to look at p53 regulated micro RNAs in HCT116 cells that were treated with Inauhzin-C (INZ-C) and identified miR-139-5p as a new micro RNA p53 target. Their data show quite convincingly that this is a real p53 target since it is induced by Doxorubicin and INZ-C in a p53-dependent manner, that the miR-139-5p gene contains a p53 binding site to which Dox-induced p53 binds (by ChIP analysis) and that this sequence can serve as a p53 response element in a luciferase reporter assay. The authors go further and use computational analysis to identify a target of miR-139-5p as cAMP-specific phosphodiesterase4D (PDE4D) and in Figure 3 show that a mimic of miR-139-5p leads reduces expression of PDE4D while and inhibitor of miR-139-5p increased the levels of PDE4D. PDE4D is known to increase conversion of cAMP to AMP and in Figure 4 the authors show increased levels of cAMP upon introduction of the miR-139-5p mimic, and corresponding increased levels of Bim, shown previously to be downstream of cAMP mediated cell death. Finally, they show results from a clinical colon cancer patient dataset indicating a reciprocal relationship between miR-139-5p and PDE4D levels.

Identifying novel target of p53 is an important area of study, especially when these targets provide new insights into the mechanism of action of p53. Thus, the focus of this study could be appropriate for *eLife* if the authors were to provide more information as outlined in the comments below. The problem with this study, however, is that the authors do not have enough information as to whether, how or when the induction of miR-139-5p by p53 actually plays a role in any p53 outcomes in cells, in animals or in human cancer. There is no evidence for this newly described process in a physiological or pathological setting in vivo. For publication in a journal, such as *eLife*, some demonstration of the importance of this process in vivo needs to be provided. The authors would need to produce extensive and convincing data to allay the above concerns as well as the following more specific comments:

1) Figure 3. What is the relative concentration of the mature endogenous miR-139-5p when induced by p53 compared to the concentration of the miR-139-5p mimic? This is important because it is not clear whether p53 induction of endogenous miR-139-5p is sufficient to cause enough of a drop in PDE4D to have any impact on cAMP.

2) Figure 3 shows that there is a massive increase in the levels of PDE4D when the miR-139-5p inhibitor in introduced into cells even without Dox treatment. One would therefore expect that the levels of cAMP would be reduced. But in Figure 4 this is not the case. They only see an impact of the inhibitor on the levels that are increased by Dox treatment. Why is this?

3) Figure 4: the authors show data indicating a modest but apparently convincing drop in cell viability upon introduction of the miR-139-5p mimic into cells. If BIM is the reason for this, then one would assume that knockdown of BIM would increase cell viability. But there does not appear to be any rescue by BIM knockdown in cells treated with the mimic and so their claim that the impact of miR-139-5p mimic is attenuated by BIM is not substantiated by the data. Understanding the biological consequences of this pathway is central to establishing this microRNA as an important target for p53. Knockdown of BIM is an important experimental approach. Multiple targeting sequences should be used for this knockdown to lessen the possibility of off-target effects. The authors also need to show that knockdown of BIM blocks some cellular outcome due to p53, presumably proliferation inhibition.

4) If p53 works via miR-139-5p to increase levels of cAMP after doxorubicin as the authors propose, they need to show this. Similarly, they need to show that the increase in BIM is p53 dependent. This is central to their conclusion.

5) While the authors show in Figure 4 an inverse correlation between miR-139-5p and PDE4D, is there a positive correlation between wild-type p53 and miR-139-5p in tumors?

6) Many of the differences shown, especially, the impact of mir-139-p on cell death and the impact of BIM knockdown on this cell death (Figure 4) are very small. This questions the importance of this newly described process.

7) It is unclear whether the authors examined the p53 status in the clinical samples. Given the focus of the manuscript on p53 regulation, one would expect that p53 loss of mutation would be associated with higher levels of PDE4D. If this is not the case, the authors should comment on this.

8) The authors also need to address whether p53 is sufficient for these effects or do they only occur with DNA damage.

9) Along these lines as well, it should be tested whether the miR-139-5p inhibitor blocks this p53-dependent cellular outcome.

10) Although the authors cite examples from the published literature, it remains to be shown what is the nature of inhibition of proliferation that is seen. Use of additional assays beyond that of MTT would be informative in this regard.

11) The biological significance of the two bands for PDE4D that are seen should be discussed.

[Editors’ note: The second decision letter after peer review is shown below.]

Thank you for resubmitting your work entitled "Inactivation of Oncogenic cAMP-specific Phosphodiesterase 4D by miR-139-5p in Response to p53 Activation" for further consideration by *eLife*. Your article has been reviewed by two peer reviewers, and the evaluation has been overseen by a Reviewing Editor and a Senior Editor. We regret to inform you that your work will not be considered further for publication in *eLife*.

While the referees indicate that your revised paper is of potential interest for *eLife*, the remaining issues are substantial, and both referees were concerned that you did not respond adequately to the previous critiques. If you can provide convincing data to address their concerns, your paper can be reconsidered for *eLife*. Alternately, as more experimentation would be required, you might want to submit your study to a more specialized journal.

Major critique:

In this revised paper the authors have responded to the most serious criticism that came from Reviewers 1 and 3, who made it clear that it did not go far enough to show physiological significance of the p53 target. The authors provide xenograft data, which are for the most part convincing and strengthen the paper. However, some of the previous concerns remain. If the authors can respond to these comments and provide better data where indicated this paper would be suitable for *eLife*. However, it should be noted that these are not new concerns, but rather points that the authors failed to properly address.

1) Most importantly, there was a concern about the biological relevance of the findings. The authors have now included a xenograft study. While this extends observations to an in vivo setting, they still have not shown that cellular outcomes are dependent upon this pathway. They state that there is Bim-mediated cell-cycle arrest and yet data was presented to argue that the outcome may in fact be apoptosis. It is not clear how BIM would induce a cell-cycle effect and the authors have not clearly stated which outcome they think is occurring in response to miR-139-5p regulation by p53. They note that BIM siRNA causes effects on cell viability and on the percent of cells with a hypodipold DNA content and yet the model in Figure 5 shows BIM causes a cell cycle effect. This is confusing.

2) As pointed out by reviewers 1 and 3 there are some cases (for example Figure 3 and Figure 4) where effects were very modest and even if statistically significant. It is difficult to reconcile such modest effects with meaningful physiological outcomes.

3) Responses to the reviewers' comments were in some cases not really satisfactory. In one case (their point 3) they combine comments by two referees and respond to them as if they were one referee's comment. To this melded comment they provide data for the reviewers' eyes that 2 different siRNAs vs BIM have the same effect on Doxorubicin suppression of H460 cell proliferation. First, the "rescue" by BIM knockdown was very modest in the H460 cells. Second, the cells they use to show this are not the same that were in the paper in Figure 4, where they used A549 cells. It does not seem very relevant to present data on a different assay in another cell line to make their point.

4) In another case their response to the comment "If p53 works via miR-139-5p to increase levels of cAMP as the authors propose, they need to show this". It was noted that cAMP levels do not track with outcomes. The authors' response was one of speculation rather than providing additional insights through experimental data.

5) Similarly, they need to show that the increase in BIM is p53 dependent. What they show is not very convincing. First, in Figure 4 doxorubicin is not really appropriate to use as a demonstration of p53-specific response because this drug has other p53-independent effects on cells, so Nutlin would be a better control. Second, in Figure 4—figure supplement 2, it is not appropriate to compare two cell lung cancer lines that differ in p53 status as a way to prove that BIM is induced in a p53-specific manner. They use wild-type p53 expressing H460 cells and compare these to p53-null H1299 cells. Yet these two cell lines differ in many more ways besides p53 status. An isogenically matched set of cells is needed to make this important point that the increase in BIM needs p53 under these conditions. Also the extent of induction is really very modest in the H460 cells. Either use the HCT116 pair of cells or knockdown p53 in the H460 cells.

Reviewer #1:

This revised study by Cao et al. has identified a miR139-5p as a gene that is transactivated by p53 in Figure 1 and 2. In Figure 3 they provide evidence that PDE4D is a target of miR139-5p. They show in Figure 4 that levels of cAMP a known target of PDE4D, are up in miR139-5p mimic-containing cells or in Doxorubicin-treated cells. Also BIM, previously shown to be upregulated by cAMP, is increased by the miR-139 mimic. Finally in new data in Figure 5, they provide data from mouse xenograft studies and show that SW480 cells harboring a construct expression miR-139-5p produce smaller tumors than cells with a control vector and a representative sample shows lower PDE4D and Ki67 staining.

In this revised paper they have responded to the most serious criticisms that came from Reviewers 1 and 3 who made it clear that it did not go far enough to show physiological significance of the p53 target. The new xenograft data are for the most part convincing and strengthen the paper. However, some of the previous concerns remain. If the authors can respond to these comments and provide better data where indicated this paper would be suitable for *eLife*.

1) As pointed out by reviewers 1 and 3 there are some cases (for example Figure 3 and Figure 4) where effects are very modest and even if statistically significant. It is difficult to reconcile such modest effects with meaningful physiological outcomes.

2) Their responses to the comments are in some cases not really satisfactory. In one case (their point 3) they combine comments by two referees and respond to them as if they were one referee's comment. To this melded comment they provide data for the reviewers' eyes that 2 different siRNAs vs BIM have the same effect on Doxorubicin suppression of H460 cell proliferation. First, the "rescue" by BIM knockdown is very modest in the H460 cells. Second, the cells they use to show this are not the same that were in the paper in Figure 4 where they used A549 cells. It does not seem very relevant to present data on a different assay in another cell line to make their point.

3) In another case their response to the comment "If p53 works via miR-139-5p to increase levels of cAMP as the authors propose, they need to show this. Similarly, they need to show that the increase in BIM is p53 dependent". Is not very convincing. First, in Figure 4. Doxorubicin is not really appropriate to use as a demonstration of p53-specific response because this drug has other p53-independent effects on cells, so Nutlin would be a better control. Second, in Figure 4—figure supplement 2 it is not appropriate to compare two cell lung cancer lines that differ in p53 status as a way to prove that BIM is induced in a p53-specific manner. Also the extent of induction is really very modest in the H460 cells. Either use the HCT116 pair of cells or knockdown p53 in the H460 cells.

Reviewer #2:

This is a revision of a previous submitted manuscript which attempts to argue that miR-139-5p is a novel p53 target and its mechanism of action is to increase cAMP levels via downregulation of phosphodiesterase 4 (PDE4D) and resulting Bim-mediated cell growth arrest. While the authors have addressed many of the concerns of the previous review, there remain several notable issues. If these can be addressed, the manuscript would be acceptable for publication. However, it should be noted that these are not new concerns, but rather points that the authors failed to properly address.

First, and most importantly, there was a concern about the biological relevance of the findings. The authors have now included a xenograft study. While this extends observations to an in vivo setting, they still have not shown that cellular outcomes are dependent upon this pathway. They state that there is Bim-mediated cell cycle arrest and yet data is presented to argue that the outcome may in fact be apoptosis. It is not clear how BIM would induce a cell cycle effect and the authors have not clearly stated which outcome they think is occurring in response to miR-139-5p regulation by p53. They note that BIM siRNA causes effects on cell viability and on the percent of cells with a hypodipold DNA content and yet the model in Figure 5 shows BIM causes a cell cycle effect. This is confusing.

Second, it was noted that cAMP levels do not track with outcomes. The authors' response is one of speculation rather than providing additional insights through experimental data.

Third, the p53-dependence of the BIM increase remains an issue. They use wild-type p53 expressing H460 cells and compare these to p53-null H1299 cells. Yet these two cell lines differ in many more ways besides p53 status. An isogenically matched set of cells is needed to make this important point that the increase in BIM needs p53 under these conditions.

---

## [Author Response]

[Editors’ note: the author responses to the first round of peer review follow.]

*[…] Identifying novel target of p53 is an important area of study, especially when these targets provide new insights into the mechanism of action of p53. Thus, the focus of this study could be appropriate for eLife if the authors were to provide more information as outlined in the comments below. The problem with this study, however, is that the authors do not have enough information as to whether, how or when the induction of miR-139-5p by p53 actually plays a role in any p53 outcomes in cells, in animals or in human cancer. There is no evidence for this newly described process in a physiological or pathological setting in vivo. For publication in a journal, such as eLife, some demonstration of the importance of this process in vivo needs to be provided.*

We thank the editor for the suggestion and agree that in vivo demonstration is critical for our finding. Therefore, we performed xenograft study using SW480 cells stably overexpressing miR-139-5p or scramble control. The data are now included in Figure 5 and interpreted in the manuscript accordingly.

*The authors would need to produce extensive and convincing data to allay the above concerns as well as the following more specific comments:*

*1) Figure 3. What is the relative concentration of the mature endogenous miR-139-5p when induced by p53 compared to the concentration of the miR-139-5p mimic? This is important because it is not clear whether p53 induction of endogenous miR-139-5p is sufficient to cause enough of a drop in PDE4D to have any impact on cAMP.*

We agree with the reviewer that the relative level of miR-139-5p is an important measure for sufficiently transducing p53-mediated effect. As shown in Figure 6 in this letter, induction of p53 by 4μM INZ-C or 0.5 μM Doxorubicin led to 6 to 8-fold increase of miR-139-5p in H460 cells (Figure 6). Transient transfection of pSIF-H1-miR-139-5p and miR-139-5p mimic in A549 cells resulted in ~4-fold and ~7800-fold increase of miR-139-5p, respectively (Figure 6). However, we were able to detect changes in PDE4D protein at comparable levels among cells treated with 4μM INZ-C (Figure 1), 0.5 μM Doxorubicin (Figure 3), or transfected with pSIF-miR-139-5p (Figure 6) or miR-139-5p mimic (Figure 3). Consistently, 0.5 μM Doxorubicin treatment and miR-139-5p mimic transfection also led to similar induction of cAMP (Figure 4). These observations suggest that although the relative levels of miR-139-5p could differ significantly when cells were treated with different means, several folds of increase achieved by p53 induction is sufficient to suppress PDE4D protein expression and subsequently cAMP elevation, while the excess amount of miR-139-5p mimic does not cause further impact.

Author response image 1.Downregulation of PDE4D by p53 and miR-139-5p.(**A**) H460 cells were treated with DMSO, 2 μM INZ-C or 0.5 μM Dox for 18 hr followed by qRT-PCR analysis of miR-139-5p. (**B**) and (**C**) H460 cells were transfected with pSIF-H1-Scramble control (SC) or pSIF-H1-miR-139-5p (pSIF-139) (**B**), or negative control oligos (Ctrl) or miR-139-5p mimic (Mimic), followed by qRT-PCR analysis of miR-139-5p at 48 hr after transfection. Samples from (**A**) and (**B**) were also analyzed by Western blot as shown in (**D**) and (**E**), respectively.**DOI:**
http://dx.doi.org/10.7554/eLife.15978.019

*2) Figure 3 shows that there is a massive increase in the levels of PDE4D when the miR-139-5p inhibitor in introduced into cells even without Dox treatment. One would therefore expect that the levels of cAMP would be reduced. But in Figure 4 this is not the case. They only see an impact of the inhibitor on the levels that are increased by Dox treatment. Why is this?*

We thank the reviewer for the comment. One possible scenario is that the basal level of PDE4D in cells is sufficient to keep cAMP at a certain level, while increase in PDE4D protein would not further reduce it.

*3) Figure 4: the authors show data indicating a modest but apparently convincing drop in cell viability upon introduction of the miR-139-5p mimic into cells. If BIM is the reason for this, then one would assume that knockdown of BIM would increase cell viability. But there does not appear to be any rescue by BIM knockdown in cells treated with the mimic and so their claim that the impact of miR-139-5p mimic is attenuated by BIM is not substantiated by the data. Understanding the biological consequences of this pathway is central to establishing this microRNA as an important target for p53. Knockdown of BIM is an important experimental approach. Multiple targeting sequences should be used for this knockdown to lessen the possibility of off-target effects. The authors also need to show that knockdown of BIM blocks some cellular outcome due to p53, presumably proliferation inhibition.*

We agree with the reviewer that BIM knockdown did not increase cell viability in our experiments. This is consistent with the reports that BIM knockdown itself has minor effect on cell apoptosis, while BIM is more responsible for upstream signal stimulation (Lin, Xu et al., 2013, Zambon, Wilderman et al., 2011). In Figure 4, we showed that in the presence of BIM siRNA, introduction of miR-139-5p mimic lost the ability to significantly decrease cell viability, indicating that BIM is responsible, at least in part, for miR-139-5p effect.

To address the reviewer’s comment on the off-target effects of BIM siRNA, we also tested another BIM siRNA in addition to the original one, and found that knocking down BIM with both of these two siRNAs reversed doxorubicin suppression of H460 cell growth determined by using the sulforhodamine B (SRB) assay as described (Vichai & Kirtikara, 2006) (Figure 7), confirming that the effect of BIM knockdown is less likely due to off-target, and also indicating BIM is a downstream effector of p53 pathway as we have proposed in this study.

Author response image 2.Knockdown of BIM rescue doxorubicin suppression of H460 cell proliferation.H460 cells were transfected with two siRNAs against BIM, and 48 hr after transfection, cells were treated with 0.5 μM Dox for indicated duration and cell proliferation was determined by using the sulforhodamine B (SRB) assay. * p<0.05. Error bars, SD.**DOI:**
http://dx.doi.org/10.7554/eLife.15978.020

4) If p53 works via miR-139-5p to increase levels of cAMP after doxorubicin as the authors propose, they need to show this. Similarly, they need to show that the increase in BIM is p53 dependent. This is central to their conclusion.

We thank the reviewer for the advice. In Figure 4, we showed that Doxorubicin increased cAMP to a level comparable to that induced by miR-139-5p mimic (Figure. 4A), while in the presence of miR-139-5p inhibitor, this induction no longer existed, indicating that Doxorubicin/p53 effect on cAMP is through miR-139-5p.

In order to address the dependence of BIM increase on p53, we treated H460 and H1299 cells, which are p53-positive and p53-negative cell lines, respectively, with 2 μM INZ-C or 10 μM mycophenolic acid (MPA, activator of p53 (Sun, Dai et al., 2008)), and found that BIM is induced by these two compounds only in H460 cells, but not H1299 cells (Figure 4—figure supplement 2), suggesting that the increase in BIM is p53 dependent.

*5) While the authors show in Figure 4 an inverse correlation between miR-139-5p and PDE4D, is there a positive correlation between wild-type p53 and miR-139-5p in tumors?*

We agree with the reviewer that p53 mutation status in the clinical samples should be associated with miR-139-5p/PDE4D levels. Unfortunately, we don’t have the first hand data on p53 status in this set of tumor samples. However, it is well accepted that p53 mutation rate in colon cancer is 50~60% according to the literature (Iacopetta, 2003, Kandoth, McLellan et al., 2013). In addition, even though p53 is wild type in some tumors, its activity should remain quite low due to other tumor favorable pathways. Therefore, it is difficult to elucidate the correlation between p53 mutation status and miR-139-5p/PDE4D levels, but we, as well as many other p53 researchers, believe that the overall p53 activity in the tumors is very low based on a large amount of literature information and the “killing” nature of p53 activity.

*6) Many of the differences shown, especially, the impact of mir-139-p on cell death and the impact of BIM knockdown on this cell death (Figure 4) are very small. This questions the importance of this newly described process.*

We agree with the reviewer that miR-139-5p did not show very dramatic impact on cell death in our in vitro data. However, the difference between control and miR-139-5p transfected cells is statistically significant. Also, previous reports showed that miR-139-5p is a potential tumor suppressor against various types of cancer (Guo, Hu et al., 2012, Wong, Wong et al., 2011), which are consistent with our findings. In addition, our in vivo data in the revised manuscript further confirmed the tumor suppressor role of miR-139-5p, and tumors overexpressing miR-139-5p have elevated expression of BIM (Figure 5), suggesting the correlation between miR-139-5p and BIM as we have proposed.

*7) It is unclear whether the authors examined the p53 status in the clinical samples. Given the focus of the manuscript on p53 regulation, one would expect that p53 loss of mutation would be associated with higher levels of PDE4D. If this is not the case, the authors should comment on this.*

See response to point #5.

*8) The authors also need to address whether p53 is sufficient for these effects or do they only occur with DNA damage.*

We thank the reviewer for the advice. To address this, we treated HCT116 p53^+/+^ and HCT116 p53^-/-^ cells with 10 μM Nutlin-3 (Nut), which disrupts MDM2-p53 interaction (Vassilev, Vu et al., 2004), or 5 nM Actinomycin D (ActD) (Iapalucci-Espinoza & Franze-Fernández, 1979), which leads to ribosomal stress-mediated p53 activation at low concentration, and found that these two drugs only reduced PDE4D in p53 positive, but not p53 negative, cells (Figure 3—figure supplement 3), indicating this reduction is p53 dependent, and not limited to DNA damage caused by Doxorubicin.

9) Along these lines as well, it should be tested whether the miR-139-5p inhibitor blocks this p53-dependent cellular outcome.

We thank the reviewer for the advice. As shown in Figure 4—figure supplement 3, introduction of miR-139-5p inhibitor significantly alleviated A549 cell growth inhibition by several p53 activating drugs, including INZ-C, Actinomycin D and Nutlin-3. The rescue effect was not dramatic, indicating that miR-139-5p is only partially responsible for p53-mediated downstream events.

10) Although the authors cite examples from the published literature, it remains to be shown what is the nature of inhibition of proliferation that is seen. Use of additional assays beyond that of MTT would be informative in this regard.

We thank the reviewer for the advice and performed flow cytometry analysis on H460 cells transfected with either pSIF-H1-scramble control or pSIF-H1-miR-139-5p. The result in Figure 8 showed that miR-139-5p transfection caused significant increase in cell population of sub G1, indicating that miR-139-5p could induce apoptosis as well.

Author response image 3.miR-139-5p induces apoptosis.H460 cells were transfected with pSIF-H1-Scramble control (Control) or pSIF-H1-miR-139-5p (139-5p), and 48 hr after transfection, the cells were subjected to flow cytometry analysis. *p<0.05. Error bars, SD.**DOI:**
http://dx.doi.org/10.7554/eLife.15978.021

*11) The biological significance of the two bands for PDE4D that are seen should be discussed.*

We thank the reviewer for the comment and now include the discussion in the revised version.

[Editors’ note: the author responses to the second round of peer review follow.]

*Reviewer #1:*

This revised study by Cao et al. has identified a miR139-5p as a gene that is transactivated by p53 in Figure 1 and 2. In Figure 3 they provide evidence that PDE4D is a target of miR139-5p. They show in Figure 4 that levels of cAMP a known target of PDE4D, are up in miR139-5p mimic-containing cells or in Doxorubicin-treated cells. Also BIM, previously shown to be upregulated by cAMP, is increased by the miR-139 mimic. Finally in new data in Figure 5, they provide data from mouse xenograft studies and show that SW480 cells harboring a construct expression miR-139-5p produce smaller tumors than cells with a control vector and a representative sample shows lower PDE4D and Ki67 staining. In this revised paper they have responded to the most serious criticisms that came from Reviewers 1 and 3 who made it clear that it did not go far enough to show physiological significance of the p53 target. The new xenograft data are for the most part convincing and strengthen the paper. However, some of the previous concerns remain. If the authors can respond to these comments and provide better data where indicated this paper would be suitable for eLife.

We thank the reviewer for his/her positive view of our manuscript. In the revised manuscript, we are providing point-by-point responses to specific concerns.

1) As pointed out by reviewers 1 and 3 there are some cases (for example Figure 3 and Figure 4) where effects are very modest and even if statistically significant. It is difficult to reconcile such modest effects with meaningful physiological outcomes.

We agree with the reviewer that in some cases, our in vitro data did not show very dramatic effects. Specifically, for Figure 3, since the pMIR-reporter assay is an ectopic luciferase system as a very sensitive and reliable method, we believe that statistically significant changes (20-35% inhibition) are meaningful. For Figure 4, we realized that in A549 cells, miR-139-5p is relatively less effective in inhibiting cell growth, so in our first revision, we included another cell line H460 and performed flow cytometry analysis, which showed quite dramatic induction of sub-G1 cell population (3, first rebuttal letter).

Additionally, our xenograft study further confirmed the correlation between miR-139-5p and PDE4D, BIM and tumor growth (Figure 5 and Figure 5—figure supplement 1). Collectively, our study demonstrates the tumor suppressor role of miR-139-5p in both in vitroand in vivosystems.

2) Their responses to the comments are in some cases not really satisfactory. In one case (their point 3) they combine comments by two referees and respond to them as if they were one referee's comment. To this melded comment they provide data for the reviewers' eyes that 2 different siRNAs vs BIM have the same effect on Doxorubicin suppression of H460 cell proliferation. First, the "rescue" by BIM knockdown is very modest in the H460 cells. Second, the cells they use to show this are not the same that were in the paper in Figure 4 where they used A549 cells. It does not seem very relevant to present data on a different assay in another cell line to make their point.

We thank the reviewer for the comment. The original decision letter was provided to us as a melded summary, so we did not have separate comments from different reviewers. That was why we tried to address point 3 as one comment. In Figure 7 in the first rebuttal letter, we showed that BIM knockdown rescued H460 cell inhibition by Doxorubicin from 58% to 42% (siBIM-1) and 35% (siBIM-2) at 48 h, and from 67% to 47% (siBIM-1) and 37% (siBIM-2) at 72 h after treatment, respectively, which we believe is quite evident. To be consistent with the data in Figure 4 as the reviewer suggested, we also performed cell proliferation assay in A549 cells to test the rescue effect of BIM knockdown in response to Nutlin-3 treatment. As shown in Figure 4—figure supplement 5, the inhibitory effect of Nutlin-3 on cell proliferation was significantly reversed by BIM siRNA. Although the rescue effect was not 100%, we believe it is reasonably anticipated, because as one of hundreds of target genes of p53, miR-139 also has multiple target genes other than PDE4D/cAMP/BIM pathway (Guo, Hu et al., 2012, Shen, Liang et al., 2012, Wong, Wong et al., 2011). Therefore, we propose that induction of miR-139-5p mediated PDE4D/cAMP/BIM pathway contributes to p53 tumor suppressor function.

3) In another case their response to the comment "If p53 works via miR-139-5p to increase levels of cAMP as the authors propose, they need to show this. Similarly, they need to show that the increase in BIM is p53 dependent". Is not very convincing. First, in Figure 4. Doxorubicin is not really appropriate to use as a demonstration of p53-specific response because this drug has other p53-independent effects on cells, so Nutlin would be a better control. Second, in Figure 4—figure supplement 2 it is not appropriate to compare two cell lung cancer lines that differ in p53 status as a way to prove that BIM is induced in a p53-specific manner. Also the extent of induction is really very modest in the H460 cells. Either use the HCT116 pair of cells or knockdown p53 in the H460 cells.

We agree with the reviewer that Nutlin-3 is a more appropriate approach to show p53 dependence. We now have replaced Figure 4 with new data showing that Nutlin-3 treatment also increased cAMP level in A549 cells, but not in the presence of miR-139-5p inhibitor.

We agree with the reviewer that comparing BIM induction in H460 and H1299 cells is not appropriate to show p53 dependence and therefore performed Western blot analysis in HCT116 p53^+/+^ and HCT116 p53^-/-^ cells, and replaced Figure 4—figure supplement 2 with new data (now Figure 4—figure supplement 4) showing that Nutlin-3 induced BIM expression only in HCT116 p53^+/+^, but not HCT116 p53^-/-^, cells.

*Reviewer #2:*

This is a revision of a previous submitted manuscript which attempts to argue that miR-139-5p is a novel p53 target and its mechanism of action is to increase cAMP levels via downregulation of phosphodiesterase 4 (PDE4D) and resulting Bim-mediated cell growth arrest. While the authors have addressed many of the concerns of the previous review, there remain several notable issues. If these can be addressed, the manuscript would be acceptable for publication. However, it should be noted that these are not new concerns, but rather points that the authors failed to properly address.

We thank the reviewer for his/her positive view of our manuscript, which is potentially acceptable upon satisfactory responses to specific concerns.

First, and most importantly, there was a concern about the biological relevance of the findings. The authors have now included a xenograft study. While this extends observations to an in vivo setting, they still have not shown that cellular outcomes are dependent upon this pathway. They state that there is Bim-mediated cell cycle arrest and yet data is presented to argue that the outcome may in fact be apoptosis. It is not clear how BIM would induce a cell cycle effect and the authors have not clearly stated which outcome they think is occurring in response to miR-139-5p regulation by p53. They note that BIM siRNA causes effects on cell viability and on the percent of cells with a hypodipold DNA content and yet the model in Figure 5 shows BIM causes a cell cycle effect. This is confusing.

We thank the reviewer for the comments. In addition to Figure 3—figure supplement 3 and Figure 4—figure supplement 3 (now Figure 4—figure supplement 6), in which we showed that Nutlin-3 treatment led to p53 specific suppression of PDE4D and miR-139-5p inhibitor significantly rescued Nutlin-3 inhibition (in addition to INZ-C and Actinomycin D) of cell proliferation, we have now included more data (Figure 4, Figure 4—figure supplement 3, Figure 4—figure supplement 4 and Figure 4—figure supplement 5), showing that specific p53 activation by Nutlin-3 induced cAMP dependent on miR-139- 5p, and cell growth inhibition by Nutlin-3 is at least partially dependent on miR-139-5p and BIM. Combined with our in vivo observation, we believe that the tumor suppressor function of p53 is at least partially through our proposed pathway.

We agree with the reviewer that BIM is a mediator of apoptosis. Indeed, in Figure 3 in the first rebuttal letter, we used sub G1 population as an indicator of apoptosis (Zhou, Hao et al., 2015). In fact, in our proposed model in Figure 5, we showed that BIM actually caused cell growth arrest as a general term of cellular outcome, instead of a cell cycle effect.

Second, it was noted that cAMP levels do not track with outcomes. The authors' response is one of speculation rather than providing additional insights through experimental data.

We thank the reviewer for the comment. We now have included Figure 4—figure supplement 3 to show that the rescue effect of miR-139-5p on A549 cell growth inhibition by Nutlin-3 is correlated with the change of cAMP levels (Figure 4).

*Third, the p53-dependence of the BIM increase remains an issue. They use wild-type p53 expressing H460 cells and compare these to p53-null H1299 cells. Yet these two cell lines differ in many more ways besides p53 status. An isogenically matched set of cells is needed to make this important point that the increase in BIM needs p53 under these conditions.*

We agree with the reviewer that H460 and H1299 cells are not an appropriate pair of cells for comparing p53 dependent BIM induction. Therefore, we performed Western blot analysis in HCT116 p53^+/+^ and HCT116 p53^-/-^ cells, and replaced Figure 4—figure supplement 2 with new data (now Figure 4—figure supplement 4) showing that Nutlin-3 induced BIM expression only in HCT116 p53^+/+^, but not in HCT116 p53^-/-^ cells.